# Resource Scheduling for Multitarget Imaging in a Distributed Netted Radar System Based on Maximum Scheduling Benefits

**DOI:** 10.3390/s22176400

**Published:** 2022-08-25

**Authors:** Tianchen Hu, Kefei Liao, Shan Ouyang, Haitao Wang

**Affiliations:** Guangxi Key Laboratory of Wireless Wideband Communication and Signal Processing, Guilin University of Electronic Technology, Guilin 541004, China

**Keywords:** resource scheduling, distributed netted radar, multitarget imaging, scheduling benefit

## Abstract

As a new radar system with improved performance, distributed multiple-input multiple-output (MIMO) radar provides a new idea for the development of netted radar. Aiming at the limited resource allocation problem of netted radar, this paper extends the idea of distributed MIMO radar to netted radar and proposes a resource scheduling algorithm for multitarget imaging in distributed netted radar based on the maximum scheduling benefits. Under the condition of the cognition of the target characteristics, the algorithm comprehensively considers the angle and dwell time to complete the multiradar and multitarget matching. Then it uses the principle of compressed sensing to calculate the pulse resources required for sparse imaging of each target on the corresponding radar. In this paper, the scheduling benefit of a radar system is expressed by weighting the three factors of the scheduling success rate, the hit value rate and the pulse resource consumption rate. The resource scheduling model is established according to the maximum scheduling benefits and solved using a heuristic algorithm. The simulation results show that compared with the traditional algorithm, this method improves the scheduling benefits of the radar system.

## 1. Introduction

With the development of science and technology and the revolution of military systems, radar application scenarios are becoming increasingly more complex. The severe air situation requires the radar system to reasonably allocate limited resources to ensure that as many tasks as possible are performed [1]. The radar system increases the adaptive ability of resource scheduling by introducing cognitive ideas [2,3], thus greatly improving the success rate of task scheduling. In [4,5], the method of sparse inverse synthetic aperture radar (ISAR) imaging based on compressed sensing theory lays a foundation for the resource scheduling of imaging in netted radar.

To date, some domestic and foreign scholars have performed a considerable amount of research on radar resource scheduling. In reference to the multifunctional phased array radar imaging task scheduling problem, Reference [6] proposes an adaptive scheduling algorithm for ISAR imaging based on sparse observation, which saves considerable pulse resources through sparse observation. Aiming at the problem of radar system resources easily reaching the upper limit, Reference [7] proposes a pulse interleaving algorithm based on time pointers, which refines the time resources to the pulse level and makes full use of the pulse resource waiting period. However, the research object of the above algorithms is a single radar.

In the conventional netted radar system, each radar adopts the self-transmitting and self-receiving working mode. In order to cope with the modern combat environment, the optimal deployment method of netted radar based on detection probability was proposed in [8]. Reference [9] comprehensively considers the system resource consumption and the overall multitarget tracking performance,; the reference proposes a novel resource management optimization model. In [10], in response to the urgent need for high-resolution imaging with limited resources, a netted radar resource scheduling method based on game theory was proposed. In [11], a time and aperture resource allocation strategy for multitarget inverse synthetic aperture radar (ISAR) imaging that incorporates requirements for distributed network radar multitarget imaging tasks is proposed to improve the efficiency of a radar system. An optimal scheduling algorithm for multi-functional networked cognitive radar resources based on maximizing scheduling benefits was proposed in [12]. In general, the research on netted radar resource scheduling mainly focuses on radar location, power allocation, target detection and tracking. In contrast, there is less research on target imaging with a more restricted resource allocation. In addition, when the radar system needs to handle a large number of tasks, the pulse resources consumed by the conventional netted radar system increase sharply, and the echoes of the targets received by the radars are likely to interfere with each other.

Space military confrontation is a brand new combat mode, whether it is offensive or defensive.Threat assessment of space targets is an important link in the process of space operations. Reference [13] analyzed the types of space targets and their threat indicators in detail, and a threat assessment model is established by using the method of threat membership and information entropy. In [14], a new method for evaluating target threat degree in air defense system is proposed. The threat factors of air targets are divided into quantitative indexes and qualitative indexes. The total threat value is obtained by weighting the threat value of each indicator by the AHP method. A comprehensive decision-making method of target threat assessment and ranking is given in Reference [15]. According to the principle of maximum membership degree, a single target optimization model is established, aiming at the intercepting target of a surface-to-air missile weapon system.

MIMO radar has many incomparable advantages over traditional phased array radar. MIMO radar usually adopts two antenna configurations, namely distributed MIMO radar and colocated MIMO radar. Reference [16] uses a novel method to accurately estimate the attributes of multiple targets by observing them from different angles to provide spatial diversity in the MIMO radar system with widely spaced antennas. A joint beam and power scheduling algorithm for multitarget tracking in MIMO radar was proposed in [17]. Reference [18] studies the power allocation of MIMO radar in target tracking. The target imaging task often requires radar to transmit many pulses to a target, which has a more limited resource allocation. Therefore, the imaging task will have higher requirements for resource scheduling and management in netted radar systems. Reference [19] proposes a new high-resolution bistatic ISAR imaging framework based on compressed sensing, and its robustness is verified by simulation. Reference [20] showed that 3D ISAR imaging usually needs a 2D synthetic aperture to provide azimuth and elevation information. Several radars are selected from the radar system to complete the target imaging, and then the 3D ISAR image reconstruction model is constructed by combining the projection relationship of each 2D ISAR image in [21].

In summary, this paper extends the idea of distributed MIMO radar to netted radar, so that each radar in the system can act as a transmission–receipt radar, a transmission radar or a receipt radar. Therefore, in the case of the same number of radars, the distributed netted radar system proposed in this paper has more signal transmission channels than the conventional netted radar system.It obtains more degrees of freedom from the resource management perspective. According to the above analysis, imaging tasks have more limited resource allocations than target tracking and detection tasks. Therefore, this paper studies the resource scheduling problem of multitarget imaging in distributed netted radar. The main contributions are as follows:(1)References [19,20,21] show that bistatic imaging can be completed by using compressed sensing, and 2D ISAR images from different observation angles can be fused into 3D ISAR images. Therefore, at least three noncollinear radars can be selected in the distributed netted radar to complete the 3D imaging of a target. According to the function value and the distance between the radar and target to complete its 2D imaging, we design a radar and target pairing function and select three noncollinear radars for each target.(2)Generally, radar resource management aims to improve performance as much as possible when the resources are limited or to reduce system consumption as much as possible when performance is satisfied. However, resource management technology can also be used to achieve a good compromise between performance and resource consumption. In this paper, we consider that the scheduling revenue of the entire radar system is related to the scheduling success rate and the hit value rate of the task, and the scheduling costs are related to the pulse resources consumed to complete the task. Therefore, the overall scheduling benefit of the radar system is the weight of the scheduling revenues and scheduling costs. Finally, this paper takes the maximum scheduling benefits of the entire radar system as the objective function. Then it takes the system resources and imaging requirements as constraints, establishes the resource scheduling model and uses a heuristic algorithm to solve the problem.

The remainder of this paper is arranged as follows. Section 2 introduces the distributed netted radar system model. Section 3 introduces the parameter analysis of the target. Section 4 introduces the resource scheduling model and solution. Section 5 gives the simulation results. Section 6 gives the conclusion.

## 2. Distributed Netted Radar System Model

Consider a netted radar system that consists of multiple radars. Each radar in the system can act as a transmission–receipt radar, a transmission radar or a receipt radar. The system generally uses the data processing center to comprehensively process the target information from different radars. Therefore, this paper establishes a distributed netted radar system model, as shown in Figure 1. It should be noted that netted radar may include various radars, but for the convenience of analysis, this paper assumes that each radar is a 2D ISAR imaging radar. Target *i* (*i* = 1, 2, …, *m*) is target *i* in the system, and Radar *j* (*j* = 1, 2, …, *n*) is radar *j* in the system.

## 3. Target Parameter Analysis

The scheduling model of distributed netted radar is established according to the important parameters such as the azimuth coherent integration time $T_{ci,j}$ of the target imaging task, the azimuth observation dimension $M_{i,j}$, the threat of target to radar $P_{i,j}$ and the pairing function of radar and target $e_{i,j}$. In order to complete the resource scheduling for imaging in distributed netted radar, the above parameters are analyzed.

Assuming that the target moves smoothly, the radar transmits a few pulses to each target and receives the target echo. The specific information of the target is obtained through a conventional algorithm: the distance $R_{i,j}$ from *Radar j* to *Target i*, the angle between the line from the radar to the target and the beam center of radar $\beta_{i,j}$, the target speed $V_{i}$ and the angle between the target velocity and radar azimuth $\theta_{i,j}$, as shown in Figure 2.

### 3.1. Azimuth Coherent Integration Time $T_{ci,j}$ and Azimuth Observation Dimension

$M_{i,j}$ According to the method proposed in [6], the target is imaged with coarse resolution, and the target azimuth projection size $S_{xi,j}$ and azimuth sparsity $K_{i,j}$ are estimated. Then the azimuth coherent integration time $T_{ci,j}$ can be expressed as
(1)$$T_{ci,j}=\frac{S_{x\_ref}}{S_{xi,j}}\cdot \frac{R_{i,j}\cdot \lambda }{V_{i,j}\cdot \cos\left(\theta_{i,j}\right)\cdot 2\rho_{ref}}$$
where $S_{x_{-}ref}$ is the reference size of the target azimuth, and $\rho_{ref}$ is the reference resolution of the azimuth.

According to the theory of compressed sensing, the observation dimension will affect the reconstructed target imaging result. Therefore, in order to reconstruct the original signal with high probability, the observation dimensions $M_{i,j}$ of *Radar j* to *Target i* need to satisfy the following inequality:(2)$$M_{i,j}\geq c_{1}\cdot K_{i,j}\cdot \ln\left(PRF\cdot T_{ci,j}\right)$$
where $c_{1}$ is a constant related to the recovery accuracy, and usually 0.5∼2, and *PRF* is the pulse repetition ratio.

### 3.2. The Threat of the Target to Radar $P_{i,j}$

According to the threat criterion of air targets, targets with short distances, high speeds and movement towards radar are more threatening in [13]. Therefore, the calculation of the target threat in this paper mainly considers the distance $R_{i,j}$ from the target to the radar, the target speed $V_{i}$ and the angle $\theta_{i,j}$ between the target velocity and the radar azimuth. Suppose that in the scheduling interval *T*, the distributed netted radar system is composed of *n* radars and needs to image *m* targets. Using the method proposed in [11], the weight of each factor in each radar is calculated, and then the weight of each factor in *n* radars is averaged to obtain the final threat weight factor. Taking *Radar j* as an example, the three factors affecting the threat of *m* targets are written in matrix form, namely,
(3)$$A_{j}={\left(a_{i,k}\right)}_{m\times 3}$$
where $a_{i,k}$ (*i* = 1, 2, …, *m*; *k* = 1, 2, 3) represents the value of the *k*th factor of *Target i*; *k* = 1, 2 and 3 respectively, represent the distance from *Target i* to *Radar j*, the speed of *Target i* and the sine value of the angle between the velocity of *Target i* and the azimuth of *Radar j*.

In order to evaluate the threat and eliminate the influence of different physical dimensions on the evaluation results, matrix A is normalized according to Equations (Equation 4) and (Equation 5), and then the weight of each factor in *Radar j* is calculated through Equation (Equation 6).
(4)$$\left\{\begin{array}{c} \mathrm{cost}\;\mathrm{index}:\bar{r_{i,k}}=\frac{a_{i,k}-\max\left(a_{i,\cdot }\right)}{\min\left(a_{i,\cdot }\right)-\max\left(a_{i,\cdot }\right)}\\ \mathrm{revenue}\;\mathrm{index}:\bar{r_{i,k}}=\frac{a_{i,k}-\min\left(a_{i,\cdot }\right)}{\max\left(a_{i,\cdot }\right)-\min\left(a_{i,\cdot }\right)} \end{array}\right.$$
(5)$$r_{i,k}=\frac{\bar{r_{i,k}}}{\sum_{x=1}^{3}\bar{r_{i,x}}}$$
where for the cost index, the greater the value of the index is, the smaller the threat, which corresponds to the distance index in this paper. For the revenue index, the greater the value of the index is, the greater the threat, which corresponds to the target speed and the sine value of the angle between the velocity of the target and the azimuth of the radar.
(6)$${\omega_{j,}}_{k}=\frac{1+\frac{\sum_{i=1}^{m}r_{i,k}\ln r_{i,k}}{\ln m}}{\sum_{x=1}^{3}\left(1+\frac{\sum_{i=1}^{m}r_{i,x}\ln r_{i,x}}{\ln m}\right)}$$

In the formula, when $r_{i,k}=0$, $\mathrm{stipulate}r_{i,k}\ln r_{i,k}=0$.

Finally, the weights of the threat factors in *n* radars are averaged to obtain $\omega_{k}$ (*k* = 1, 2, 3) and $\omega_{1}$ + $\omega_{2}$ + $\omega_{3}$ = 1.
(7)$$\omega_{k}=\frac{\sum_{j=1}^{n}\omega_{j,k}}{n}$$

In conclusion, the threat of *Target i* to *Radar j* can be defined as
(8)$$P_{i,j}=\omega_{1}\cdot \frac{1/{R_{i}}_{,j}}{\max(1/{R_{i}}_{,j})}+\omega_{2}\cdot \frac{V_{i}}{\max(V_{i})}+\omega_{3}\cdot \frac{\sin(\theta_{i,j})}{\max(\sin\left(\theta_{{{}_{i}}_{,j}}\right))}$$

### 3.3. The Pairing Function of Radar and Target $e_{i,j}$

The imaging effect of different radars on the same target may be different. For example, the RCS of some targets may be weak or even unable to be imaged at certain observation angles. When facing this type of situation, according to the results of target feature recognition, we need to eliminate the inferior radar in the specific environment and then select the high-quality radars to complete imaging of the target, which completes the pairing of the radars and target. According to the principles of radar resource scheduling, the detection accuracy of radar is inversely proportional to the angle (between the line from the radar to the target and the beam center of the radar); the longer the dwell time of the target in the radar detection range is, the more detection information the radar can obtain. Therefore, in order to ensure the continuous and stable imaging of the target, this paper considers the minimum angle and the maximum dwell time in multiradar and multitarget assignment. According to the ISAR imaging requirements, only when the target has a rotational component relative to the radar, can 2D imaging be performed. So the dwell time factor is limited to the azimuth dwell time factor. In summary, the pairing function of Radar *j* and Target *i* is defined as
(9)$$e_{i,j}=\frac{1}{\beta_{ij}}\cdot T_{ci,j}$$
where $\beta_{i,j}$ is the angle between the line from the radar to the target and the beam center of the radar, and $T_{ci,j}$ is the azimuth coherent integration time.

In this paper, the basic characteristic parameters of the target are obtained using feature recognition. The azimuth coherent integration time and the azimuth sparse observation dimension are calculated using coarse resolution imaging. Then, the threat of each target to all radars in the radar system is obtained by the degree of membership and information entropy. Finally, we design the pairing function of radars and targets to facilitate the selection of the transmitting radar and receiving radar. The calculation of the above parameters is a necessary preparation step for resource scheduling in distributed netted radar.

## 4. Scheduling Algorithm

### 4.1. Selection of Transmitting Radar and Receiving Radar

The imaging of a single radar is 2D, and the imaging effect is directly proportional to the value of the pairing function. It is known that 2D images from different angles can be used to obtain a 3D image of a target through image information fusion. Therefore, the radar system can select three noncollinear radars according to the value of the pairing function from large to small and can complete a 3D image of a target. The conventional netted radar system adopts self-transmission and self-receipt, so we only consider the pairing of radars and targets. However, for a distributed netted radar system, it is necessary not only to consider the pairing of radars and targets but also to consider the selection of transmitting radars and receiving radars. Considering that the pulse signal transmitted by radar is actually an electromagnetic wave, the loss of electromagnetic waves during their propagation is positively correlated with the distance. The shorter the signal transmission distance is, the smaller the loss. In addition, the observation signal transmitted by a radar close to a target can decrease the time delay of each radar, so that the target information contained in the echo is more accurate. Therefore, according to the value of the pairing function, the algorithm in this paper selects three radars for each target to complete ISAR sparse imaging and then selects the radar closest to the target from the three selected radars as the transmission radar.

### 4.2. Performance Index

Suppose that the distributed netted radar system is composed of *n* radars, and the total number of targets that the system needs to schedule is *m*. In order to verify the effectiveness of the algorithm, the following three performance indexes are defined:

(1) Scheduling success rate (SSR): The SSR is the rate of the number of targets that actually complete a 3D image of the target to the number of targets that apply for the imaging tasks within the scheduling interval *T*. It is defined as:(10)$$\mathrm{SSR}=\frac{m^{\prime }}{m}$$
where *m*${}^{\prime }$ is the number of targets that actually completed the 3-D imaging.

(2) Hit value rate (HVR): The HVR is the sum of the target’s threat of each radar actually completing the 2D imaging tasks in scheduling interval *T* compared with the sum of the target’s threat of the applied imaging tasks. It is defined as:(11)$$\mathrm{HVR}=\frac{\sum_{i=1}^{m}\sum_{j=1}^{n}X{}_{ri,j}^{\prime }\cdot P_{i,j}}{\sum_{i=1}^{m}\sum_{j=1}^{n}X_{ri,j}\cdot P_{i,j}}$$
where $X_{ri,j}$ is the receipt radar matrix selected by the system. An element value of one means that *Target i* is assigned to the imaging of *Radar j*; however, an element value of zero means that the target is not assigned to the imaging of *Radar j*. $X_{ri,j}^{{}^{\prime }}$ is the matrix that radar actually completes the imaging task. An element value of one means that *Radar j* has completed the imaging of *Target i* whereas the element value of zero means that *Radar j* has not completed the imaging of *Target i*.

(3) Pulse resource consumption rate (PRCR): In scheduling interval *T*, the PRCR is the sum of the number of transmitted and received pulses for completing the imaging task compared with the total number of transmitted and received pulses in scheduling interval *T*. It is defined as:(12)$$\mathrm{PRCR}=\frac{\sum_{i=1}^{m}\sum_{j=1}^{n}X_{ti,j}\cdot M_{i,j}+\sum_{i=1}^{m}\sum_{j=1}^{n}X_{ri,j}\cdot M_{i,j}}{2n\cdot T\cdot \mathrm{PRF}}$$
where $X_{ti,j}$ is the transmission radar matrix selected by the system, $M_{i,j}$ is the azimuth observation dimensions of *Radar j* to *Target i*, *T* is the scheduling interval, and *PRF* is the pulse repetition frequency.

If the imaging time resource allocation model is established directly according to the above three performance indexes, it will face the multiobjective function planning problem. In order to reduce the complexity of the model, the idea of the analytic hierarchy process (AHP) can be used to classify the performance indexes and transform them into the problem of single objective function planning. Therefore, this paper defines the scheduling benefit of the system and takes the maximum scheduling benefit as the objective function.

### 4.3. Scheduling Benefits of the System

To classify the above three performance indexes, the SSR and the HVR are both revenue indexes, while the PRCR is a cost index. Then, the scheduling benefits of the system (SBS) are obtained by weighting the two types of indexes using the AHP. It is defined as:(13)$$\mathrm{SBS}=\omega_{a}\cdot \mathrm{SSR}+\omega_{b}\cdot \mathrm{HVR}-\omega_{c}\cdot \mathrm{PRCR}$$

In the formula, the values of $\omega_{a}$, $\omega_{b}$, and $\omega_{c}$ can be obtained by the AHP, and $\omega_{a}$ + $\omega_{b}$ + $\omega_{c}$ = 1.

The AHP is a qualitative and quantitative multicriteria decision analysis method proposed by Professor Saaty. In this paper, in order to make the pairwise comparison of the evaluation criteria easier to operate, the numbers 1–9 and their reciprocals are used as scales. The definitions of performance indexes mentioned earlier show that the SSR refers to the completion of the system’s 3D imaging of the target, and the HVR refers to the completion of the system’s 2D imaging of the target. Therefore, there is a certain inclusive relationship between the two performance indexes. In addition, in many resource scheduling algorithms, only the maximum scheduling benefits are considered, as in [12]. Some literature considers using the least radar resources while completing the most imaging tasks. Therefore, they consider that the cost indexes are relatively less important. Therefore, this paper considers that the SSR is slightly more important than the HVR and that the HVR is as important as the PRCR. Under this analysis, we obtain the judgment matrix and determine that the judgment matrix meets the consistency test to obtain the weight of each index.

### 4.4. Algorithm Model

Taking the maximum scheduling benefit of the system as the objective function and taking the system resources and imaging requirements as constraints, the resource scheduling model of imaging in distributed netted radar based on the maximum scheduling benefits is established as follows:(14)$$\begin{array}{cc} \; & \max\left\{\omega_{a}*\frac{m^{\prime }}{m}+\omega_{b}*\frac{\sum_{i=1}^{m}\sum_{j=1}^{n}X{}_{ri,j}^{\prime }\cdot P_{i,j}}{\sum_{i=1}^{m}\sum_{j=1}^{n}X_{ri,j}\cdot P_{i,j}}-\omega_{c}*\frac{\sum_{i=1}^{m}\sum_{j=1}^{n}X_{ti,j}\cdot M_{i,j}+\sum_{i=1}^{m}\sum_{j=1}^{n}X_{ri,j}\cdot M_{i,j}}{2n\cdot T\cdot PRF}\right\}\\ \; & s.t.\left\{\begin{array}{c} X_{ti,j}\in \left\{0,1\right\}\;\mathrm{and}\;X_{ri,j}\in \left\{0,1\right\}\\ \forall j,k\in \left[1,n\right],e_{i,j}>e_{i,k};X_{ri,j}\geq {X_{r}}_{i,k}\\ \forall i\in \left[1,m\right],\sum_{j=1}^{n}X_{ri,j}=3,\sum_{j=1}^{n}X_{ti,j}=1\\ t_{sj}\leq X_{tj,j}t_{i,j}\leq t_{sj}+T-T_{ci,j}\\ \forall i,s\in \left[1,m\right],\mathrm{if}X_{\mathrm{ti}j}P_{i,j}>X_{tsj}P_{s,j};X_{\mathrm{ti}j}t_{i,j}<X_{tsj}t_{s,j}\\ \forall j\in \left[1,n\right],\sum_{i=1}^{m}X_{\mathrm{ri}j}M_{i,j}<T\cdot PRF \end{array}\right. \end{array}$$ The three selected radars are not collinear, and the transmitting radar is closest to the target, where the objective function is to maximize the scheduling benefits of the radar system. Here, *m* is the total number of targets applying for imaging, and $m{}^{\prime }$ is the number of targets actually completing 3D images of the targets. $X_{ti,j}$ is the transmission radar matrix selected by the system, and $X_{ri,j}$ is the receipt radar matrix selected by the system. $X{{}^{\prime }}_{ri,j}$ is the matrix in which the radar actually completes the imaging task. In the first constraint, if the matrix element value is one, *Target i* will be allocated to *Radar j* for transmission or receipt. If the matrix element value is zero, *Target i* will not be assigned to *Radar j* for imaging scheduling. The second constraint means that the radar with the largest value of the pairing function is selected first. The third constraint means that each imaging task adopts a transmission radar and three receipt radars for imaging. The fourth constraint means that the initial observation time of *Target i* for *Radar j* is within the scheduling interval, and the imaging task can be completed within the scheduling interval. The fifth constraint means that the target with the highest threat is scheduled first. The sixth constraint condition means that the pulse resources used for any radar scheduling are less than the total resources that the radar can provide.

### 4.5. Model Solving

The above model is an NP-hard problem with equality and inequality constraints. A heuristic algorithm can be used to solve the optimization problem and obtain feasible solutions. The specific steps are as follows:

The flowchart of the proposed algorithm is shown in Figure 3.

Step 1: The *N* radars in the system transmit a small amount of pulses to all the targets in the detection area and receive the target echoes. The relevant parameters of the targets are obtained through coarse resolution imaging. These parameters include the distance $R_{i,j}$ from *Target i* to *Radar j*, the speed $V_{i}$ of *Target i*, the angle $\theta_{i,j}$ between the velocity of *Target i* and the azimuth of *Radar j* and the angle $\beta_{i,j}$ between the line from the radar to the target and the beam center of the radar. We combine Equation (Equation 9) to complete the selection of radar transceiver sites and the allocation of radar front resources.

Step 2: Let the initial Radar *j* = 1.

Step 3: Perform scheduling preprocessing on *Radar j*; let the number of beams formed by the *Radar j* be $B_{j}$; let beam *k* = 1.

Step 4: Perform scheduling preprocessing on beam *k*, and assume that the number of targets with imaging requirements of beam *k* in a single scheduling interval is $M_{k}$. Then $M_{k}$ targets are added to the application list in the order of threat, and the task *i* = 1.

Step 5: Take out the *Target i* in sequence, and take the most forward time among the remaining idle times in the current scheduling interval as the initial observation time $t_{i,j,k}$ of the target. Then the final observation time $T_{ci,j}$+$t_{i,j,k}$ of the target is calculated according to the effective azimuth dwell time $T_{ci,j}$ of the target.

Step 6: Determine whether the request of *Target i* satisfies the constraint conditions of the array resources and the time resources. If the constraints and imaging requirements are met, $M_{i,j}$-2 observation pulses are randomly inserted between the start and end observation times of the *Target i*, making the current target *i* = *i* + 1. If it is not satisfied, we directly set the current target *i* = *i* + 1.

Step 7: First, determine whether the request sequence of the task has completed the traversal. If $i\leq M_{k}$, return to Step 5; if $i>M_{k}$, let *k* = *k* +1. Then determine whether the beam has completed the traversal. If $k\leq B_{j}$, return to Step 4; if $k>B_{j}$, let *j* = *j* +1. Finally, determine whether the network access radar has completed the traversal. If j≤N, return to Step 3; otherwise, the processing of this scheduling is ended.

## 5. Simulation Results

It is assumed that the radars in the distributed netted radar system are synchronized in time and that all transmit chirp signals. In addition to the different positions of the targets in the target group, other characteristic parameters are the same, such as speed, size, etc. The transmission signal carrier frequency is $f_{c}$ = 10 GHz, pulse width Tp = 1 us, bandwidth *B* = 300 MHz, pulse repetition frequency *PRF* = 1000 Hz and scheduling interval *T* = 1 s. There are 4 radars and 6 targets in scenario 1 and 6 radars and 20 targets in scenario 2, as shown in Figure 4 and Figure 5, respectively.

First, the target in Scenario 1 is recognized, and the target parameters are calculated by the method of Section 2. Then, according to the pairing function value of the radars and the targets, three radars are selected for each target for imaging. Then the transmission radar is selected from the three radars according to the distance between the target and radars. Finally, the distance between the target and the selected radar, the coherent integration time of the selected radar for target imaging and the threat of the target to the radar are obtained.

According to Table 1, three radars are selected for each target according to the value of the pairing function of radars and targets from large to small; that is, each target is assigned to the three radars with the best imaging effect in the system. Target 1 and Target 2 are assigned to Radar 1, Radar 2 and Radar 4. Target 3 and Target 4 are assigned to Radar 2, Radar 3 and Radar 4. Target 5 and Target 6 are assigned to Radar 1, Radar 3 and Radar 4.

According to Table 2, among the three radars with the best imaging effect of the target, a radar transmission observation pulse is selected for each target according to the principle of the minimum distance from the target to the radar. Target 1, Target 2 and Target 5 are assigned to Radar 1 for observation. Target 3 and Target 4 are assigned to Radar 2 for observation. Target 6 is assigned to Radar 4 for observation.

Table 2, Table 3 and Table 4 show the parameters after the radar selection is completed. According to the parameters in the table and the heuristic algorithm, we can obtain the resource scheduling results of the above six targets of the distributed netted radar system. The resource scheduling transmission timing diagram and receiving timing diagram of the target task on the corresponding radar are shown in Figure 6 and Figure 7, respectively.

Figure 6a is the result of the resource scheduling transmission timing diagram of all radars in a scheduling interval. The diagram shows that Radar 3 does not transmit observation pulses to any target. Combining Figure 6a,b verifies that the observation order of the targets in each radar is consistent with the threat order of each target to the corresponding radar.

Figure 7 is the result of the resource scheduling receiving timing diagram of all radars in a scheduling interval. The diagram shows that Radar 1 completes the scheduling of Target 1, Target 2, Target 5 and Target 6. The number of pulses received by Radar 1 to Target 1–6 is 132, 130, 0, 0, 129 and 130. Radar 2 completes the scheduling of Target 1, Target 2, Target 3 and Target 4. The number of pulses received by Radar 2 to Target 1–6 is 132, 130, 117, 118, 0 and 0. However, Radar 3 only completes the scheduling of Target 3, Target 4 and Target 5 and does not schedule Target 6. The number of pulses received by Radar 3 to Target 1–6 is 0, 0, 117, 118, 129 and 0. Radar 4 completes the scheduling of Target 3, Target 4, Target 5 and Target 6 but does not schedule Target 1 and Target 2. The number of pulses received by Radar 4 to target 1–6 is 0, 0, 117, 118, 129 and 130. Radar 3 does not schedule Target 6 because after Radar 3 completes the scheduling of Target 5, Target 4 and Target 3 with high threats. The remaining observation time cannot meet the azimuth coherent integration time of Target 6. Therefore, Target 6 with a low threat of Radar 3 will be abandoned; that is, it will not be imaged. In the same way, Radar 4 also abandons Target 1 and Target 2 with low threats. Additionally, Figure 7 shows that the scheduling of each target for each radar is sparsely distributed. Therefore, each radar in the distributed netted radar system can perform sparse imaging of the targets through the compressed sensing method to save pulse resources.

In the same scenario, the resource scheduling of multitarget imaging in the conventional netted radar system will have the same transmission timing diagram of all radars in the scheduling interval as shown in Figure 7. Comparing the resource scheduling timing diagrams in Figure 6a and Figure 7 shows that the distributed netted radar system is obviously looser than the conventional netted radar system, which will also be verified in the subsequent performance analysis.

In order to verify the effectiveness of the algorithm, we will compare the three performance indexes proposed in Section 4. This paper analyzes the conventional netted radar scheduling algorithm (referred to as the traditional algorithm) in [11]. The conventional netted radar scheduling algorithm uses a pairing function (referred to as the target allocation algorithm) and the distributed netted radar scheduling algorithm (referred to as the distributed algorithm) proposed in this paper. The traditional algorithm uses the projection size to complete radar and target pairing. The target allocation algorithm uses a pairing function to complete radar and target pairing. The distributed algorithm uses a pairing function to complete radar and target pairing. The system scheduling benefits of the three algorithms are calculated using Equation (Equation 13). The following is a simulation for Scenario 2. Figure 8 shows the comparison results of the performance indexes of the three algorithms under different numbers of target after 1000 Monte Carlo experiments are averaged.

The targets in Scenario 2 belong to three different target groups: Target Group 1, Target Group 2 and Target Group 3 that contain 6, 7 and 7 targets, respectively. Therefore, the performance index comparison results in Figure 8 are not smooth for Target 6 and Target 7. Figure 8a shows that the distributed algorithm has a higher scheduling success rate than the other two scheduling algorithms. In addition, the target allocation algorithm that uses the radar and target pairing function has a much higher scheduling success rate than the traditional algorithm, which indicates that the radar and target pairing method based on the pairing function is more reasonable than the radar and target pairing method based on the projection size. Considering that some targets cannot complete 3D imaging after system resource scheduling but can complete 2D imaging of targets using radar, the hit value rate index is defined. Figure 8b shows that the distributed algorithm proposed in this paper still slightly outperforms the other two algorithms, indicating that the sum of the target threats completed by the distributed algorithm accounts for a greater proportion of the sum of the target threats applied for imaging tasks. That is, the proposed algorithm can complete the tasks assigned by the system more efficiently. Regarding the radar system, the traditional algorithm and the target allocation algorithm should consume the same resources for imaging the same target, but the traditional algorithm in Figure 8c has a lower pulse resource consumption rate than the target allocation algorithm. This is because the scheduling success rate and hit value rate of the traditional algorithm are far lower than those of the target allocation algorithm. Nevertheless, the pulse resource consumption rate of the distributed algorithm is still the lowest. Figure 8d confirms that the distributed algorithm can complete more imaging tasks with fewer pulse resources, thereby improving the working efficiency of the radar and obtaining greater system scheduling benefits.

## 6. Conclusions

In this paper, we first studied the resource scheduling of imaging in a netted cognitive radar system based on the projection size. We find that this method ignores the 2D imaging condition when pairing radars and targets; that is, the target must have a rotational component relative to the radar. Therefore, this paper considers the angle and dwell time and designs a pairing function for radars and targets. Considering that the pulse resources consumed by the conventional netted radar increase sharply when the number of targets is large, the echoes of the targets received by each radar easily interfere with each other. Therefore, we extend the idea of distributed MIMO radar to the netted radar system and propose a resource scheduling algorithm for multitarget imaging in a distributed netted radar system based on the maximum scheduling benefits. We establish and solve the scheduling optimization model. The simulation results show that the algorithm outperforms the traditional algorithm in scheduling success rate and hit value rate. Furthermore, the algorithm also saves the transmission pulse and improves the scheduling benefit of the radar system.

## Figures and Tables

**Figure 1 sensors-22-06400-f001:**
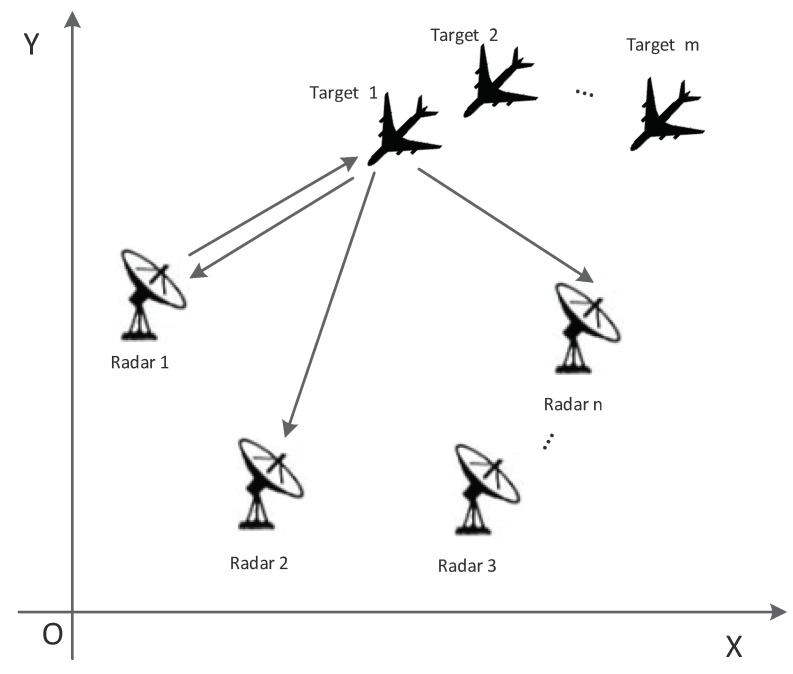
Two-dimensional position of target relative to radar in the system.

**Figure 2 sensors-22-06400-f002:**
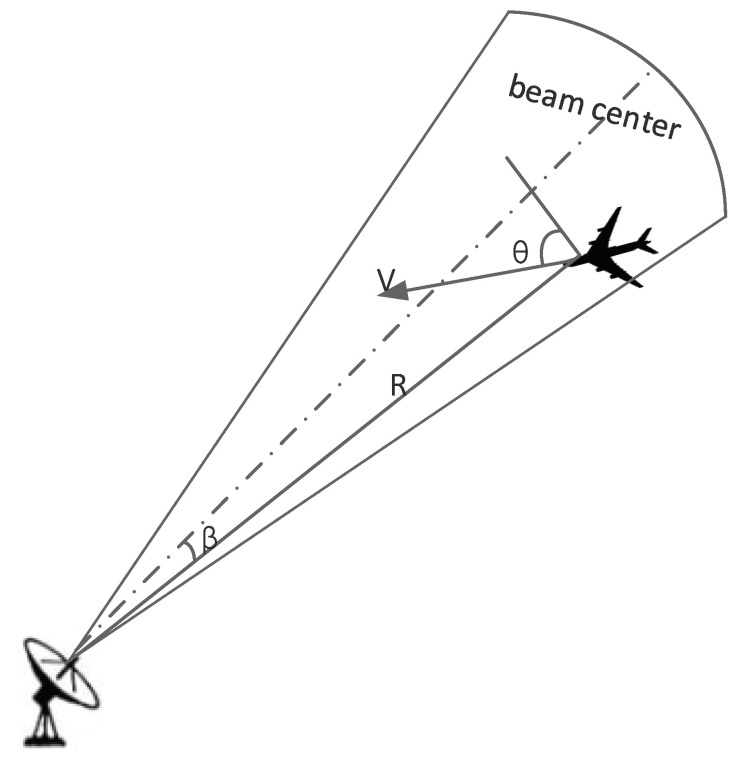
Cognitive results of target characteristics.

**Figure 3 sensors-22-06400-f003:**
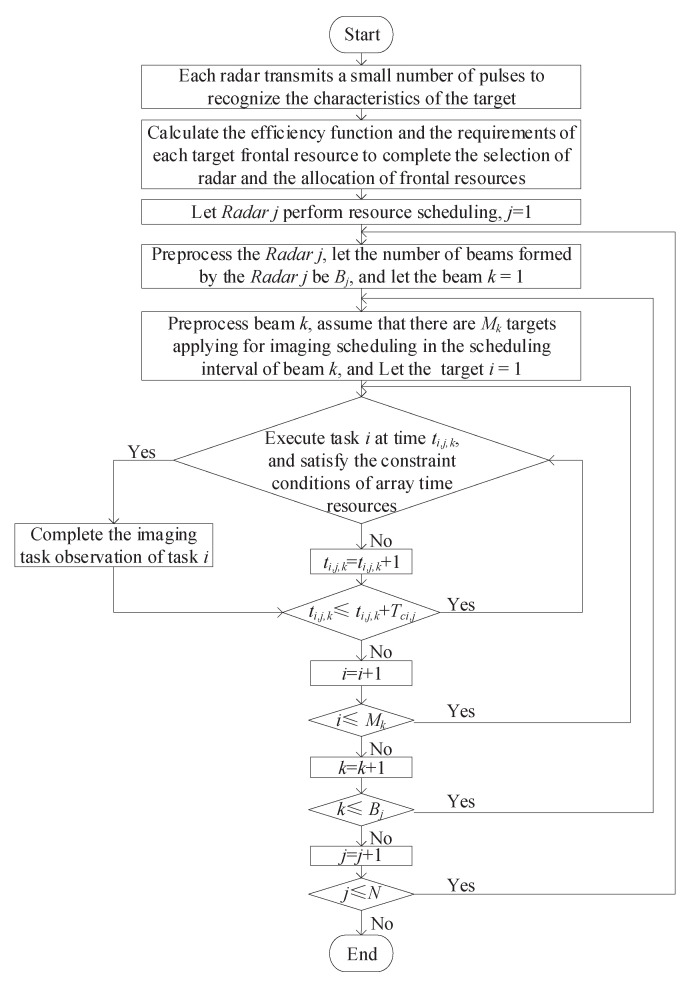
Flowchart of the proposed algorithm.

**Figure 4 sensors-22-06400-f004:**
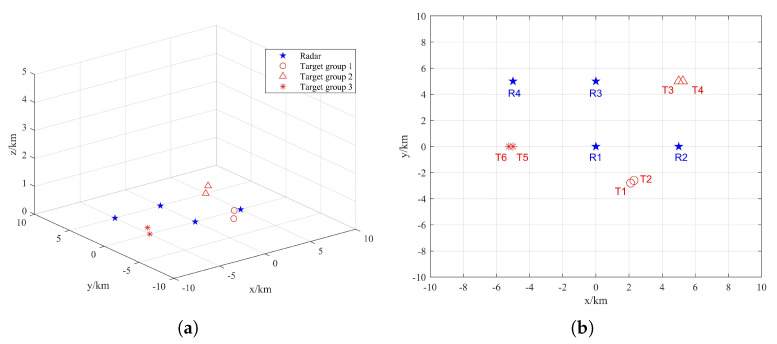
Structure distribution of radar position and target position in Scenario 1: (**a**) three dimensional view; (**b**) vertical view.

**Figure 5 sensors-22-06400-f005:**
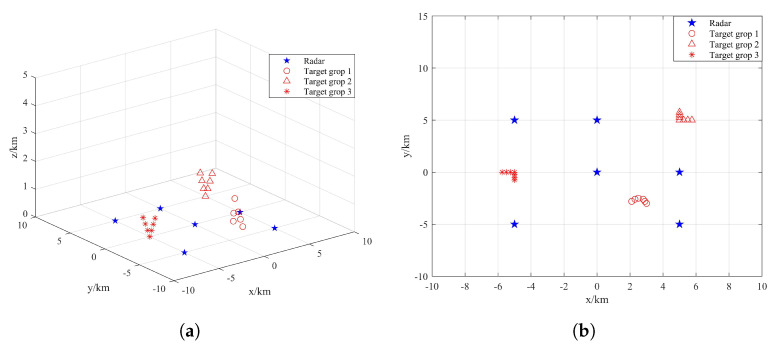
Structure distribution of radar position and target position in Scenario 2: (**a**) three dimensional view; (**b**) vertical view.

**Figure 6 sensors-22-06400-f006:**
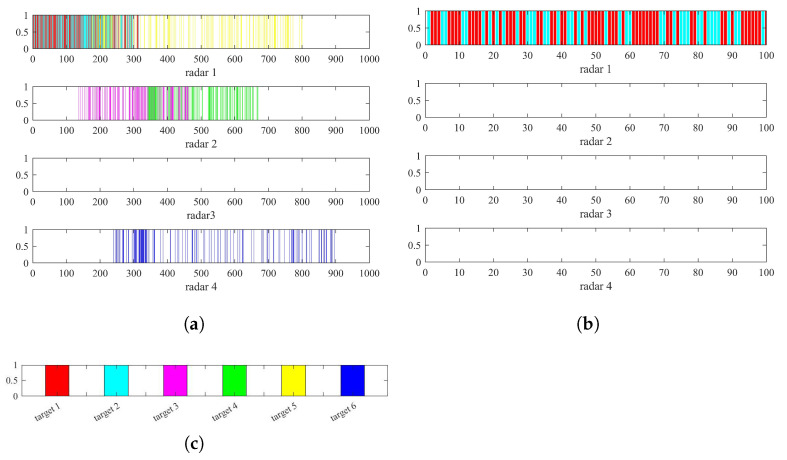
Resource scheduling transmission timing diagram of imaging: (**a**) transmission timing diagram of all radars in a scheduling interval; (**b**) transmission timing diagram of all radars in the first 100 ms; (**c**) the color corresponding to each target.

**Figure 7 sensors-22-06400-f007:**
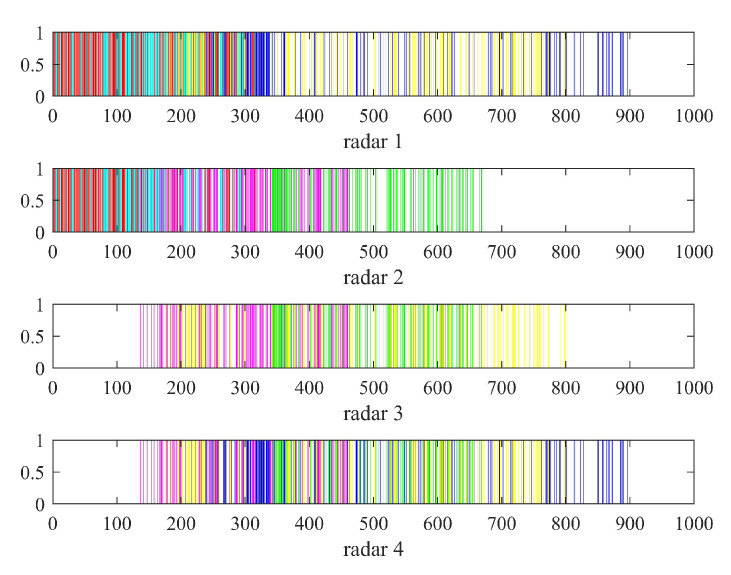
Resource scheduling receiving timing diagram of imaging.

**Figure 8 sensors-22-06400-f008:**
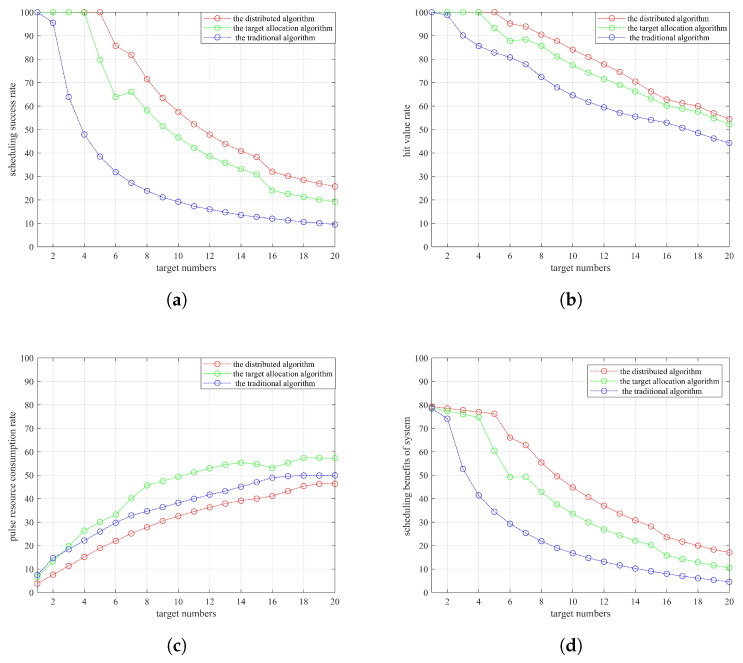
Resource scheduling transmission timing diagram of imaging: (**a**) scheduling success rate; (**b**) hit value rate; (**c**) pulse resource consumption rate; (**d**) scheduling benefit system.

**Table 1 sensors-22-06400-t001:** Pairing function of radar and target.

	Radar 1	Radar 2	Radar 3	Radar 4
Target 1	1.7951	1.9395	1.3062	1.5044
Target 2	1.7939	1.8163	1.2857	1.5044
Target 3	0	1.8701	1.9481	2.1999
Target 4	0	1.8747	2.0478	2.2556
Target 5	3.5810	1.7112	5.0644	2.0672
Target 6	3.7643	1.7544	5.1956	2.0724

**Table 2 sensors-22-06400-t002:** Distance from target to radar (unit: km).

	Radar 1	Radar 2	Radar 3	Radar 4
Target 1	3.5089	4.0389	0	10.5505
Target 2	3.5071	3.7815	0	10.5499
Target 3	0	5	5	10
Target 4	0	5.0125	5.2559	10.2530
Target 5	5	0	7.0711	5
Target 6	5.2559	0	7.2543	5.0125

**Table 3 sensors-22-06400-t003:** Coherent integration time of target in radar (unit: s).

	Radar 1	Radar 2	Radar 3	Radar 4
Target 1	0.3133	0.3385	0	0.7877
Target 2	0.3131	0.3170	0	0.7877
Target 3	0	0.3264	0.3400	0.9599
Target 4	0	0.3272	0.3574	0.9842
Target 5	0.6250	0	0.8839	0.3608
Target 6	0.6570	0	0.9068	0.3617

**Table 4 sensors-22-06400-t004:** Threat of target to radar.

	Radar 1	Radar 2	Radar 3	Radar 4
Target 1	0.9371	0.8768	0	0.5172
Target 2	0.9373	0.8987	0	0.5172
Target 3	0	0.8163	0.8163	0.6865
Target 4	0	0.8157	0.8037	0.6833
Target 5	0.7683	0	0.6923	0.6768
Target 6	0.7557	0	0.6877	0.6762

## Data Availability

The study did not report any data.

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
