# Peer review of "Resource Scheduling for Multitarget Imaging in a Distributed Netted Radar System Based on Maximum Scheduling Benefits"

_sensors, 2022, doi:10.3390/s22176400_

Round 1

Reviewer 1 Report

The resources of radar include bandwidth resources, power resources, time resources, etc. The development of resource scheduling of multi-target imaging in a distributed netted radar based on the maximum scheduling benefit in this paper, is aimed at the scheduling of time resources.

I have some questions:

1. Is the resource scheduling method  proposed in the paper suitable for all radar resources?

2. Why the paper takes 2-D ISAR radar as an example to carry out research? The working system of the radar is also a resource.

Based on the above reasons, it is suggested that the author should focus on the research of time resource scheduling based on 2-D ISAR imaging radar.

Reviewer 2 Report

An interesting and comprehensive paper on MIMO/netted radar. Well-written and backed with references and with an excellent mathematical foundation.

The only point to add, is this radar available in the real world? Authors should mentioned whether it is present in some form of defense engine or not.

Round 2

Reviewer 1 Report

In this paper, the authors present a resource scheduling method based on distributed netted radar system, which has good performance than the traditional algorithm in scheduling success rate, hit value rate and the scheduling benefit, especially in the scenario of multiple targets. 

I have a two suggestions

1. Figure 7 gives resource scheduling receiving timing diagram of imaging, the picture is more vivid, but it is not easy for readers to understand. It is better to give receiving pulse numbers of the each target with the radar.

2.  The description of  "experimental simulation" can be changed to  "simulation".
